# Recognition of Corrosion State of Water Pipe Inner Wall Based on SMA-SVM under RF Feature Selection

**Qian Zhao** [1,2], **Lu Li** [1,*], **Lihua Zhang** [1] **and Man Zhao** [3]

1    School of Communication and Information Engineering, Xi'an University of Science and Technology, Xi'an 710054, China
2    Xi'an Key Laboratory of Network Convergence Communication, Xi'an 710054, China
3    Xi'an Dishan Vision Technology Limited Company, Xi'an 712044, China
*    Correspondence: 20207223093@stu.xust.edu.cn

**Abstract:** To solve the problem of low detection accuracy of water supply pipeline internal wall damage, a random forest algorithm with simplified features and a slime mold optimization support vector machine detection method was proposed. Firstly, the color statistical characteristics, gray level co-occurrence matrix, and gray level run length matrix features of the pipeline image are extracted for multi-feature fusion. The contribution of the fused features is analyzed using the feature simplified random forest algorithm, and the feature set with the strongest feature expression ability is selected for classification and recognition. The global search ability of the slime mold optimization algorithm is used to find the optimal kernel function parameters and penalty factors of the support vector machine model. Finally, the optimal parameters are applied to support the vector machine model for classification prediction. The experimental results show that the recognition accuracy of the classification model proposed in this paper reaches 94.710% on the data sets of different corrosion forms on the inner wall of the pipeline. Compared with the traditional Support Vector Machines (SVM) classification model, the SVM model based on differential pollination optimization, the SVM model based on particle swarm optimization, and the back propagation (BP) neural network classification model, it is improved by 4.786%, 3.023%, 4.030%, and 0.503% respectively.

**Keywords:** pipeline inspection; random forest; feature selection; slime mold optimization algorithm; support vector machine



## 1. Introduction

As a key piece of equipment in modern urban construction, cast iron has good fatigue resistance and vibration reduction, so it is often used in automobile parts manufacturing, railway, machinery manufacturing, and other fields. Because of its high strength, it is also widely used in urban water supply systems. According to the American Water Association (AWWA), the vast majority of water supply pipes in the United States are gray iron pipes and ductile iron pipes, and more than 90% of the existing water supply pipelines in China use metal pipes. Still, at the same time, cast iron pipes are generally prone to rust, which can easily cause damage to the surface layer of the metal pipeline, thus resulting in a reduction in the service life of the pipeline and an increase in construction and maintenance costs [1]. To ensure safe operation and to avoid water pollution and waste of resources due to internal damage, timely and accurate pipeline damage detection is of great value for industrial applications. Corrosion exists in all areas of life, not only in water supply pipelines but also in tooth corrosion and ceramic corrosion [2–4].

Nowadays, popular pipeline defect detection methods include the magnetic flux leakage method [5], ultrasonic detection method [6], etc. However, these methods have certain limitations. The cost of detection is too high, and the area that can be detected is limited. It is difficult to detect some small areas of corrosion. Therefore, a pipeline defect detection method with wide application scope and low cost is needed [7].

In recent years, digital image processing and machine vision techniques have been rapidly developed in the field of structural health monitoring and can be effectively used to investigate defects on the external surface of pipes or other metal surfaces [8], such as corrosion and cracks. Kuo [9] et al. constructed a rust identification model based on the statistical properties of image color and the K-means clustering method, which is suitable for images with uneven illumination, However, when the image surface is uneven or the corrosion area is large or very deep, the probability of correct detection of this algorithm is low. Medeiros [10] proposed a model for classifying corroded and non-corroded surfaces using texture descriptors obtained from greyscale co-occurrence matrices and image color features. Safari and Shoorehdeli [11] applied artificial neural networks, Gabor filters, and entropy. Bondada [12] et al. detected and quantitatively assessed pipeline corrosion damage by calculating the average of image pixel saturation values. Hoang [13] proposed a method to automatically detect the corrosion of the inner wall of the water supply pipe. By combining the image texture feature extraction algorithm and the support vector machine classifier with the differential pollination optimization, the detection accuracy of the inner wall of the water supply pipe was 92.81%. Qu, ZH [14] et al. proposes a method to detect pitting corrosion by combining feature extraction and random forest algorithms, without studying more corrosion types. Nhat-Duc [15] proposed a LSHADE meta-heuristic algorithm to optimize the SVM model to detect pitting on the surface of components, with an accuracy of 91.80%, the accuracy rate needs to be improved. However, the above method only detects the presence or absence of corrosion on the pipeline without classifying and identifying different corrosion patterns, which lacks practicality and accuracy for realistic water supply pipelines with different corrosion types.

Therefore, this paper uses a combination of multiple image feature extraction and selection and Support Vector Machines (SVM) to classify and identify different corrosion patterns of pipes. The existing SVM research and applications mainly use Principal Component Analysis (PCA) methods to reduce the dimensionality of the dataset. Still, the PCA-extracted principal components have a certain degree of ambiguity. They are not as complete as the original samples. At the same time, the Random Forest (RF) algorithm is an excellent solution to this problem. Retaining Rahman [16] used the Random Forest algorithm to calculate and rank the feature importance, and after selecting the top-ranked features, used SVM to classify the proteins. However, the random forest algorithm for feature selection suffers from the problem of not considering the impact of correlation between feature variables on recognition accuracy, so this paper uses feature simplification (FS) to reduce the effect of redundant features on the random forest algorithm. In the face of high-dimensional feature data, the feature simplification algorithm can improve the performance of the random forest algorithm in feature selection, further enhancing the timeliness of the algorithm and the accuracy of subsequent recognition.

In addition, SVM is highly dependent on determining parameters such as kernel function parameters and penalty factors, so optimizing the optimal parameters is the key to improving the generalization ability of SVM models. The Particle Swarm Optimization (PSO) algorithm, a population intelligence-based stochastic search algorithm, is commonly used to optimize the kernel function parameters and penalty factors of SVM models. Li. F [17] proposed a PSO-SVM-based method for predicting the probability of failure of pressure pipelines. Although the PSO algorithm can optimize the parameters of the SVM model, the PSO algorithm itself lacks stochasticity and quickly falls into the dilemma of local optimum. In this paper, the Slime Mold Algorithm (SMA) [18] is proposed to optimize the parameters of the support vector machine classification model. The Slime Mould Optimisation algorithm has the advantages of solid convergence performance, few tuning parameters, and easy operation, and it can maintain a balance between local optimality search and global search, which can meet the needs of optimizing the internal parameters of support vector machines in this paper.

Therefore, this paper combines the image feature extraction and selection algorithm, as well as the support vector machine classification model to achieve the classification and

recognition of pipeline inner wall corrosion, and uses the feature simplified random forest (RF) algorithm to improve the related algorithm to improve the performance of random forest algorithm feature selection. The Slime Mold algorithm (SMA) is used to optimize the parameters in the SVM model to build the SMA-SVM classification model. Finally, the model is applied to the data set of pipe wall corrosion to classify and identify the damage to the pipe's inner wall.

## 2. Related Work

### 2.1. Video Image Acquisition of Pipeline Corrosion

In this paper, the industrial endoscope video capture platform built and assembled by ourselves is used to obtain the video image of pipeline inner wall corrosion [19], the video capture platform is shown in Figure 1, and Figure 2 is a sample image of the video after processing.

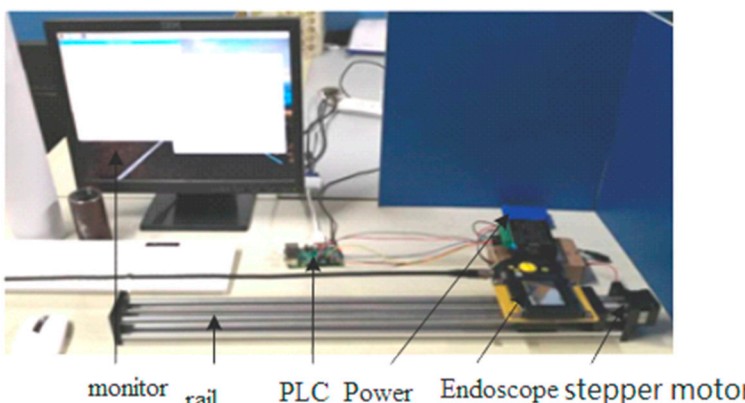

**Figure 1.** Video image acquisition platform.

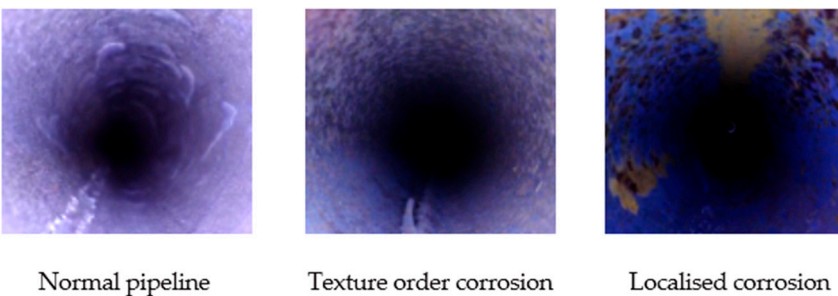

**Figure 2.** Partial image acquisition result map.

The image obtained using the above acquisition platform contains 3D information. In order to facilitate the subsequent image feature extraction, it is necessary to conduct panoramic expansion of the 3D image. As the pipeline is usually buried underground or in a dark environment, the acquired image is dark and the corrosion area is not obvious in the background, so an image enhancement algorithm is required to preprocess the image. Figure 3 is the process diagram of pretreatment.

The pre-processing process of the original image is as follows: first, use the cone-based bidirectional projection model to expand the three-dimensional image into a two-dimensional image [20]; secondly, the improved Retinex algorithm of bidirectional illumination estimation model is used to improve the image contrast and enhance the brightness of the corroded area [21].

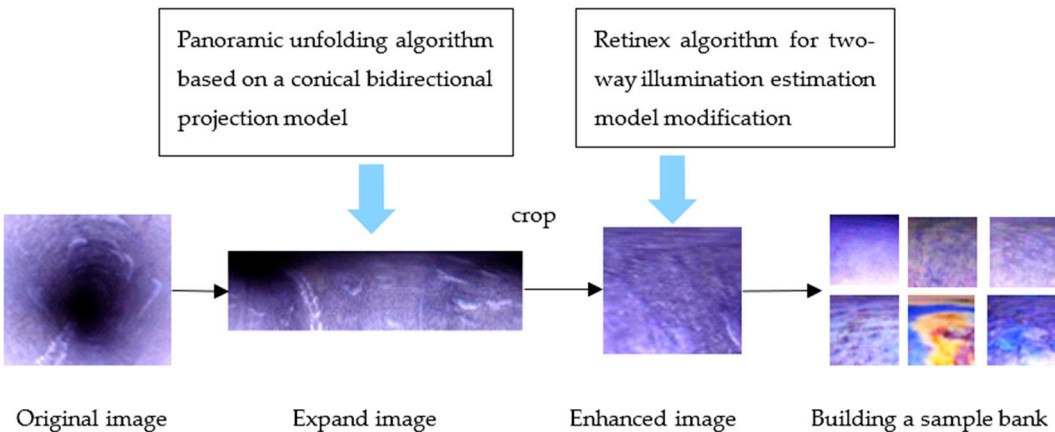

**Figure 3.** Image pre-processing process diagram.

Finally, the establishment of the pipe damage image sample library is divided into normal pipe, color sequence corrosion, texture six types of level corrosion, pitting corrosion, local corrosion, and global corrosion, as shown in Figure 4. There are 1320 sample images in the data sample library, which are stored in JPG format, including 923 training sets and 397 test sets, in order to lay the foundation for subsequent image feature extraction and recognition.

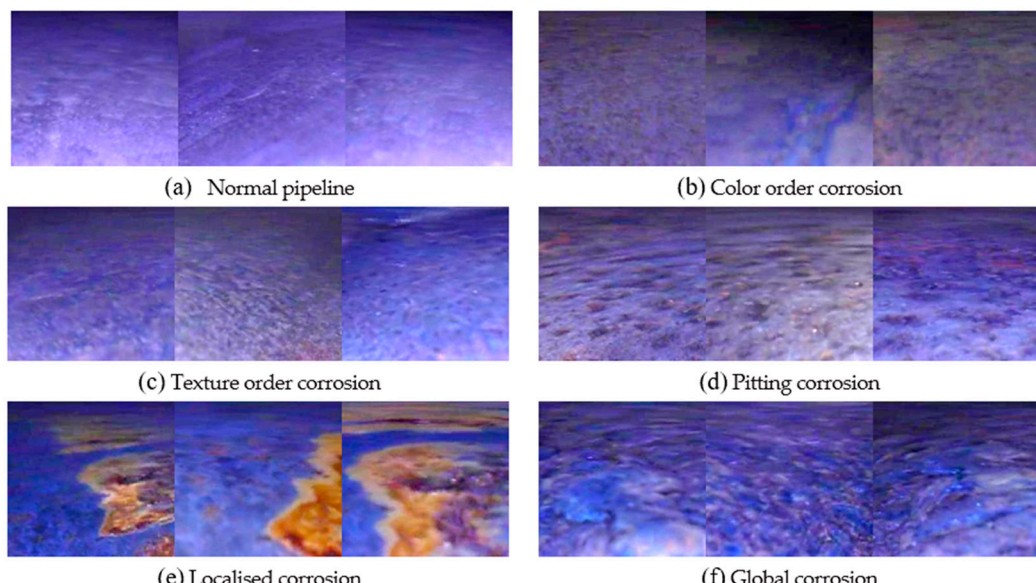

**Figure 4.** Selected images from the sample library. (**a**) Normal pipeline; (**b**) Color order corrosion; (**c**) Texture order corrosion; (**d**) Pitting corrosion; (**e**) Localised corrosion; (**f**) Global corrosion.

### 2.2. Image Feature Extraction

As the surface of the pipe contains various irregularities and its features are complex, objects with similarities to the target, such as dirt and paint, and pixels with the same color value can belong to different levels of corrosion images at the same time. Therefore, the information provided by just one pixel is not sufficient for corrosion detection. Thus, in this paper, multiple features are extracted from the corrosion image of the pipe's inner wall, including color features, greyscale co-occurrence matrix, and greyscale travel matrix features [22]. The features are fused and selected to form a useful feature set for subsequent damage type identification. Color features [23]: This paper uses the statistical properties of the image samples' three color channels (red, green, and blue) to represent the image features. The mean, standard deviation, skewness, kurtosis, color entropy, and color range of the image color moments are extracted to characterize the color statistics of the pipeline

image. R. Haralick developed the Gray Level Co-Occurrence Matrix (GLCM) as a texture feature in 1973 by studying the spatial correlation characteristics of image greyscales [24]. In this paper, the eigenvalues of the four edge parameters (second-order angular moments, contrast, correlation, entropy) of the Gray-Occurrence Matrix are extracted in four directions: 0°, 45°, 90° and 135°. The Gray-Level Run Lengths Matrix (GLRLM) is a texture description method proposed by Galloway [25]. This paper extracts 11 statistical features from the Gray-Level Run Lengths Matrix at 0°, 45°, 90°, and 135° to describe the texture statistics of a pipeline image, which can effectively identify textures of different finenesses. Based on the above feature extraction, a 78-dimensional feature dataset is constructed. The feature extraction results are shown in Tables 1–3.

**Table 1.** Extraction Results of Statistical Characteristics of Some Samples Based on Color Channels.

| Pipeline Category | | Mean Value | Standard Deviation | Skewness | Kurtosis | Color Entropy | Color Range |
|---|---|---|---|---|---|---|---|
| Normal pipe | R | 107.844 | 23.941 | 0.292 | 1.990 | 6.428 | 117 |
| | G | 97.574 | 23.492 | 0.308 | 1.939 | 6.374 | 113 |
| | B | 167.188 | 30.549 | 0.003 | 1.972 | 6.856 | 155 |
| Color order corrosion | R | 84.769 | 16.854 | 0.197 | 2.739 | 6.028 | 221 |
| | G | 80.407 | 16.747 | 0.132 | 2.853 | 6.022 | 225 |
| | B | 150.145 | 29.435 | 0.201 | 2.113 | 6.814 | 172 |
| Texture order corrosion | R | 83.570 | 11.230 | −0.771 | 3.215 | 5.387 | 70 |
| | G | 74.736 | 11.643 | −0.447 | 3.068 | 5.489 | 77 |
| | B | 136.067 | 35.168 | −0.026 | 1.832 | 6.991 | 158 |
| Pitting corrosion | R | 100.661 | 21.139 | 0.376 | 2.698 | 6.386 | 123 |
| | G | 93.763 | 20.932 | 0.391 | 2.553 | 6.359 | 115 |
| | B | 158.628 | 31.468 | −0.246 | 2.203 | 6.909 | 151 |
| Localized corrosion | R | 80.059 | 16.561 | 1.416 | 5.681 | 5.866 | 121 |
| | G | 80.037 | 13.787 | 0.804 | 3.274 | 5.707 | 91 |
| | B | 164.378 | 23.403 | −0.174 | 2.888 | 6.571 | 152 |
| Global corrosion | R | 98.441 | 14.020 | 0.476 | 3.199 | 5.813 | 94 |
| | G | 89.284 | 12.502 | 0.409 | 3.186 | 5.663 | 86 |
| | B | 142.005 | 21.896 | 0.353 | 2.506 | 6.427 | 125 |

**Table 2.** Statistical characteristics extraction results of some samples based on gray level co-occurrence matrix.

| Pipeline Category | | Angular Second Moment | Contrast Ratio | Relevance | Entropy |
|---|---|---|---|---|---|
| Normal pipe | 0° | 0.160 | 0.083 | 0.413 | 2.086 |
| | 45° | 0.145 | 0.136 | 0.410 | 2.230 |
| | 90° | 0.150 | 0.115 | 0.411 | 2.179 |
| | 135° | 0.141 | 0.151 | 0.408 | 2.258 |
| Color order corrosion | 0° | 0.190 | 0.145 | 0.756 | 1.970 |
| | 45° | 0.1529 | 0.271 | 0.719 | 2.212 |
| | 90° | 0.159 | 0.240 | 0.728 | 2.168 |
| | 135° | 0.147 | 0.292 | 0.712 | 2.249 |
| Texture order corrosion | 0° | 0.313 | 0.032 | 1.220 | 1.428 |
| | 45° | 0.296 | 0.069 | 1.202 | 1.549 |
| | 90° | 0.302 | 0.057 | 1.213 | 1.510 |
| | 135° | 0.295 | 0.070 | 1.201 | 1.553 |
| Pitting corrosion | 0° | 0.175 | 0.073 | 0.510 | 2.020 |
| | 45° | 0.153 | 0.141 | 0.505 | 2.205 |
| | 90° | 0.160 | 0.119 | 0.508 | 2.149 |
| | 135° | 0.153 | 0.145 | 0.505 | 2.212 |

**Table 2.** *Cont.*

| Pipeline Category | | Angular Second Moment | Contrast Ratio | Relevance | Entropy |
|---|---|---|---|---|---|
| Localized corrosion | 0° | 0.281 | 0.068 | 1.170 | 1.561 |
| | 45° | 0.250 | 0.132 | 1.128 | 1.730 |
| | 90° | 0.261 | 0.109 | 1.142 | 1.675 |
| | 135° | 0.252 | 0.130 | 1.130 | 1.724 |
| Global corrosion | 0° | 0.279 | 0.070 | 1.187 | 1.589 |
| | 45° | 0.242 | 0.143 | 1.134 | 1.779 |
| | 90° | 0.254 | 0.115 | 1.151 | 1.717 |
| | 135° | 0.244 | 0.137 | 1.138 | 1.768 |

**Table 3.** Extraction results of statistical properties of some samples based on gray run matrix.

| Parameter | | Normal Pipe | Color Order Corrosion | Texture Order Corrosion | Pitting Corrosion | Localized Corrosion | Global Corrosion |
|---|---|---|---|---|---|---|---|
| SRE | 0° | 0.771 | 0.857 | 0.621 | 0.809 | 0.832 | 0.809 |
| | 45° | 0.865 | 0.929 | 0.786 | 0.867 | 0.915 | 0.914 |
| | 90° | 0.833 | 0.918 | 0.737 | 0.854 | 0.897 | 0.904 |
| | 135° | 0.849 | 0.921 | 0.783 | 0.875 | 0.908 | 0.911 |
| LRE | 0° | 6.071 | 4.328 | 17.270 | 8.522 | 4.039 | 4.758 |
| | 45° | 2.106 | 0.929 | 3.062 | 2.134 | 1.574 | 1.486 |
| | 90° | 2.783 | 1.747 | 4.522 | 2.604 | 1.845 | 1.730 |
| | 135° | 2.256 | 1.554 | 3.133 | 2.017 | 1.597 | 1.507 |
| GLN | 0° | 442.623 | 740.262 | 599.922 | 454.300 | 681.941 | 746.097 |
| | 45° | 581.45 | 886.416 | 947.395 | 554.823 | 877.198 | 975.340 |
| | 90° | 543.257 | 862.205 | 823.412 | 543.484 | 848.442 | 943.703 |
| | 135° | 563.991 | 874.832 | 939.930 | 563.330 | 869.975 | 971.063 |
| RLN | 0° | 19,966.020 | 29,580.394 | 8621.792 | 21,687.847 | 27,213.109 | 24,183.847 |
| | 45° | 33,794.335 | 44,607.519 | 24,049.713 | 33,965.463 | 42,129.079 | 42,296.771 |
| | 90° | 28,945.779 | 42,181.717 | 18,916.917 | 31,661.123 | 38,679.191 | 39,982.996 |
| | 135° | 31,665.173 | 43,093.962 | 23,667.970 | 35,348.021 | 40,958.871 | 41,829.793 |
| RP | 0° | 1.197 | 0.767 | 1.097 | 1.023 | 1.341 | 1.355 |
| | 45° | 1.588 | 0.967 | 1.971 | 1.391 | 1.683 | 1.813 |
| | 90° | 1.476 | 0.938 | 1.756 | 1.338 | 1.618 | 1.755 |
| | 135° | 1.550 | 0.953 | 1.955 | 1.415 | 1.668 | 1.805 |
| LGRE | 0° | 0.002 | 0 | 0 | 0.001 | 0 | 0 |
| | 45° | 0.003 | 0 | 0.001 | 0.002 | 0 | 0 |
| | 90° | 0.003 | 0 | 0.001 | 0.002 | 0 | 0 |
| | 135° | 0.003 | 0 | 0.001 | 0.002 | 0 | 0 |
| HGRE | 0° | 3172.112 | 4325.982 | 2265.700 | 4565.655 | 2957.301 | 2633.387 |
| | 45° | 2730.462 | 4058.447 | 1998.330 | 3892.727 | 2832.746 | 2383.500 |
| | 90° | 2767.959 | 4070.036 | 1944.681 | 3897.827 | 2839.865 | 2364.071 |
| | 135° | 2752.042 | 4070.046 | 1986.400 | 3846.712 | 2827.241 | 2383.808 |
| SRLGE | 0° | 0.001 | 0 | 0 | 0 | 0 | 0 |
| | 45° | 0.002 | 0 | 0 | 0.001 | 0.001 | 0 |
| | 90° | 0.002 | 0 | 0 | 0.001 | 0.001 | 0 |
| | 135° | 0.002 | 0 | 0 | 0.001 | 0.001 | 0 |
| SRHGE | 0° | 2707.915 | 3768.246 | 1461.829 | 3965.357 | 2513.504 | 2210.679 |
| | 45° | 2487.672 | 3798.357 | 1573.591 | 3566.641 | 2614.016 | 2184.739 |
| | 90° | 2438.457 | 3762.410 | 1412.859 | 3514.164 | 2570.794 | 2134.102 |
| | 135° | 2476.439 | 3782.280 | 1551.406 | 3538.903 | 2587.578 | 2180.085 |

**Table 3.** *Cont.*

| Parameter | | Normal Pipe | Color Order Corrosion | Texture Order Corrosion | Pitting Corrosion | Localized Corrosion | Global Corrosion |
|---|---|---|---|---|---|---|---|
| LRLGE | 0° | 1 | 1 | 1 | 1 | 1 | 1 |
| | 45° | 1 | 1 | 1 | 1 | 1 | 1 |
| | 90° | 1 | 1 | 1 | 1 | 1 | 1 |
| | 135° | 1 | 1 | 1 | 1 | 1 | 1 |
| LRHGE | 0° | 7971.153 | 12,460.391 | 27,913.795 | 14,437.773 | 9478.739 | 8410.301 |
| | 45° | 4120.572 | 5516.019 | 5875.481 | 5900.670 | 4126.334 | 3485.919 |
| | 90° | 5105.659 | 6274.054 | 9086.611 | 6978.469 | 4725.028 | 4119.797 |
| | 135° | 4357.937 | 5729.299 | 6077.917 | 5732.068 | 4229.234 | 3540.333 |

### 2.3. Surface Roughness Measurement

Common methods of surface roughness measurement include probe profiler, scanning tunneling microscope (STM, R ü schlikon, Zurich, Switzerland), atomic force microscope (AFM, Bruker Corporation, Billerica, MA, USA) and some optical measurement techniques [26]. Because AFM can give a high-resolution image of the surface morphology at the atomic scale, it has the advantages of not harming the measured surface and high accuracy, AFM has brought great progress to the measurement research in this field.

In this study, RMS (root mean square, also known as $R_q$) and $R_a$ (absolute arithmetic mean) are used to quantitatively describe the surface roughness.They are calculated according to the height values of the data points in the AFM image (set the average height of each data point to 0) using the following statistical method [11], where $h_i$ is the measured surface height value and n is the number of surface height values to be counted.

$$RMS = \sqrt{\frac{1}{n}\sum_{i=1}^{n} h_i^2} \tag{1}$$

$$R_a = \frac{1}{n}\sum_{i=1}^{n} |h_i| \tag{2}$$

The AFM image of pipeline corrosion image is shown in Figure 5, and the relationship curve between *RMS* and $R_a$ values of surface roughness and AFM scanning scale is shown in Figures 6 and 7.

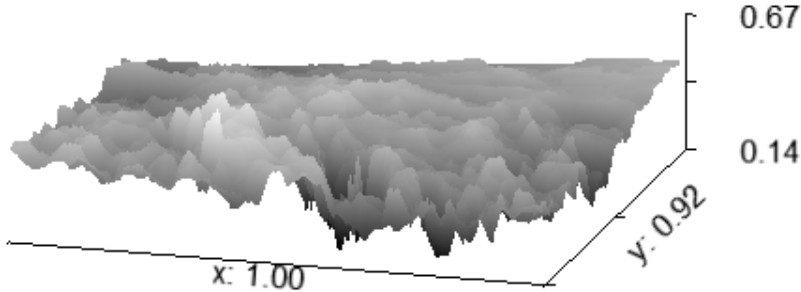

**Figure 5.** AFM Image.

It can be seen from the above figure that the surface of the image studied in this paper is rough and undulating, and there is some corrosion. Therefore, it is necessary to carry out follow-up research to classify and identify the corrosion morphology of the inner wall of the pipeline.

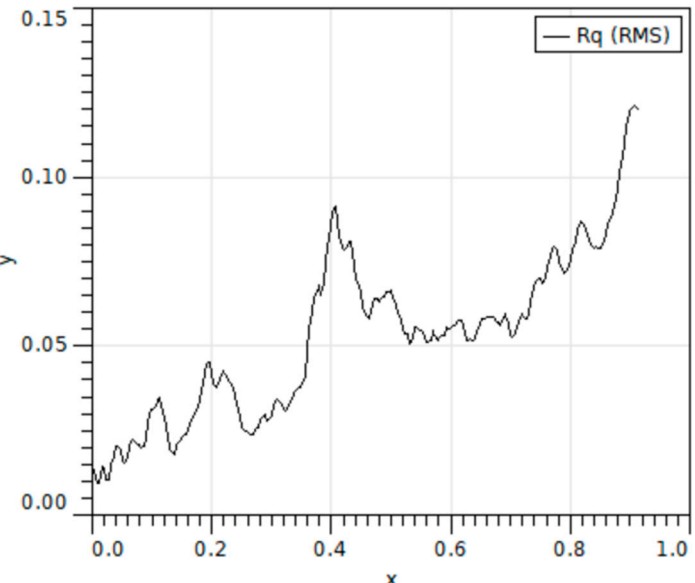

**Figure 6.** Relationship between RMS value and AFM scanning scale.

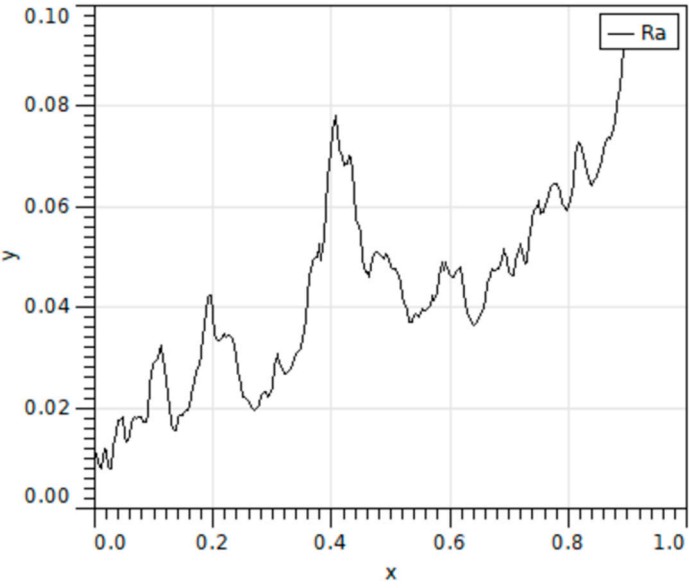

**Figure 7.** Relationship between $R_a$ value and AFM scanning scale.

### 3. Method

In this paper, a variety of defect features, including color features, gray level co-occurrence matrix, and gray level run length matrix features, are extracted from the pipeline inner wall corrosion image, and then these features are fused and combined with the random forest algorithm (FS-RF) improved by feature simplification algorithm to filter the fused features, and finally, the key and effective feature data set is extracted. Finally, the SVM classification model is used to test the extracted feature dataset, and the SMA (slime mold optimization algorithm) is used to optimize the SVM classifier, which improves the accuracy of recognition classification. The overall experimental process of this paper is shown in Figure 8.

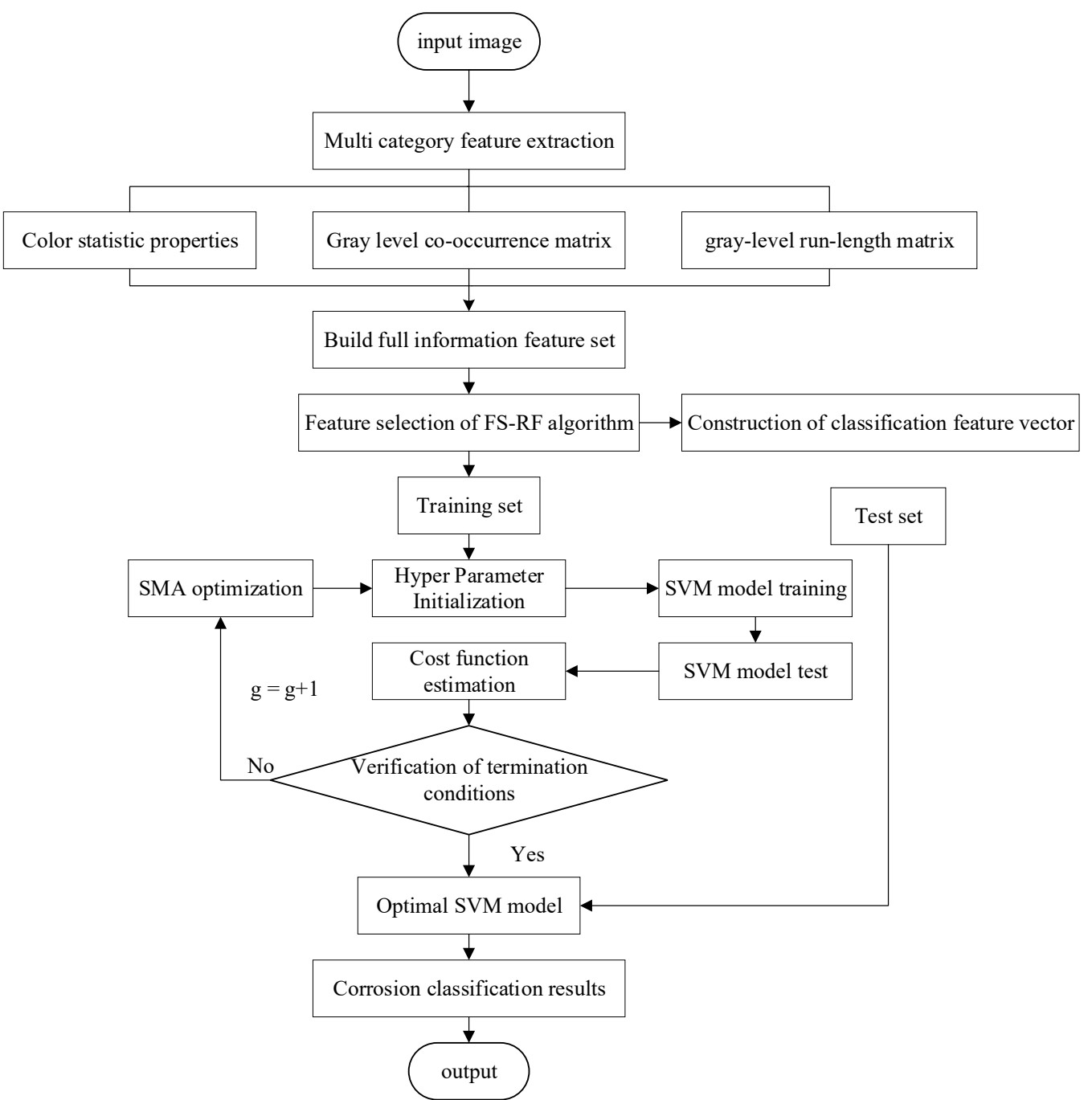

**Figure 8.** Overall flow chart of the experiment.

### 3.1. Feature Selection

In recent years, Random Forest has been widely used as a generalization method. Because it can handle many high-dimensional features, can determine the importance and relationship of features, and has no tendency to overmatch, this paper selects data features based on a random forest algorithm.

### 3.1.1. Random Forest Algorithm

In 2001, Breiman proposed the Random Forest algorithm [27], a classifier that can provide training and integrated estimation of samples using multiple decision trees. Figure 9 is the schematic diagram of its algorithm.

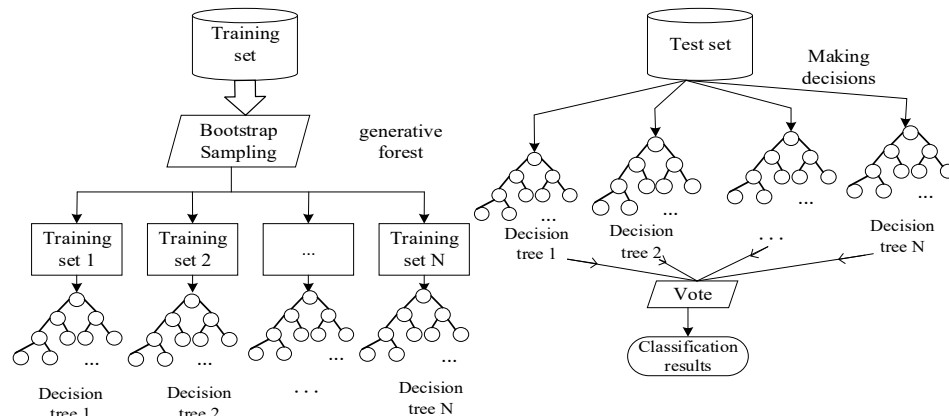

**Figure 9.** Schematic diagram of the random forest algorithm.

In a classification decision tree, the Gini index (Gini impurity) is used as a criterion for selecting features, and the Gini index for each node is calculated as

$$Gini(t) = 1 - \sum_{j=1}^{N} [p(j|t)]^2 \tag{3}$$

where $p(j|t)$ it indicates the probability that the category $j$ is selected at the node $t$, the sample data at the node $t$ is identified as the same type when the Gini impurity is zero, the smaller the value, the lower the probability that the selected samples in the set are classified wrong, i.e., the higher the purity of the set, the more information is obtained.

3.1.2. Feature-Simplified Random Forest Algorithm

The purpose of the feature simplification algorithm is to reduce the interference of some useless features of the random forest algorithm in the calculation of the Gini index value in the decision tree and to eliminate features that have little influence on the identification of damage to the inner wall of the pipe. By calculating the correlation between each feature parameter and different damage category samples, each feature parameter is assigned a corresponding weight, and the features are ranked and filtered according to the magnitude of the weights. Figure 10 is the algorithm flow chart of FS.

Step 1   A sample $R$ is randomly drawn from the candidate set of pipeline damage features, with samples from the same class $S$ being adjacent and randomly drawn from a different class of samples.

Step 2   Select one of the three samples $R\ S$ which $D$ is a unique feature $A_i$.

Step 3   Calculate the Euclidean distance between the feature $A_i$ of the sample $R$ and the feature $A_i$ of the sample $S$, denoted as $d(R_{A_i}, S_{A_i})$, and the Euclidean distance between the feature $A_i$ of the sample $R$ and the feature $A_i$ of the sample $D$, denoted as $d(R_{A_i}, D_{A_i})$.

Step 4   Repeat steps 1 to 3 and calculate the weights for each feature $A_i$ with the following formula.

$$W(A_i) = W(A_i) - \left(\frac{d(R_{A_i}, S_{A_i})}{m} - \frac{d(R_{A_i}, D_{A_i})}{m}\right)(i = 1, 2, 3, \ldots, N) \tag{4}$$

where $W(A_i)$ is the weight of the feature $A_i$, and the initial value of each feature weight is assigned to 0; $m$ is the number of repetitions; $N$ is the total number of pipe damage features, the values are assigned to $0.7 * P$ and $P$ is the total number of samples in the pipe damage feature dataset.

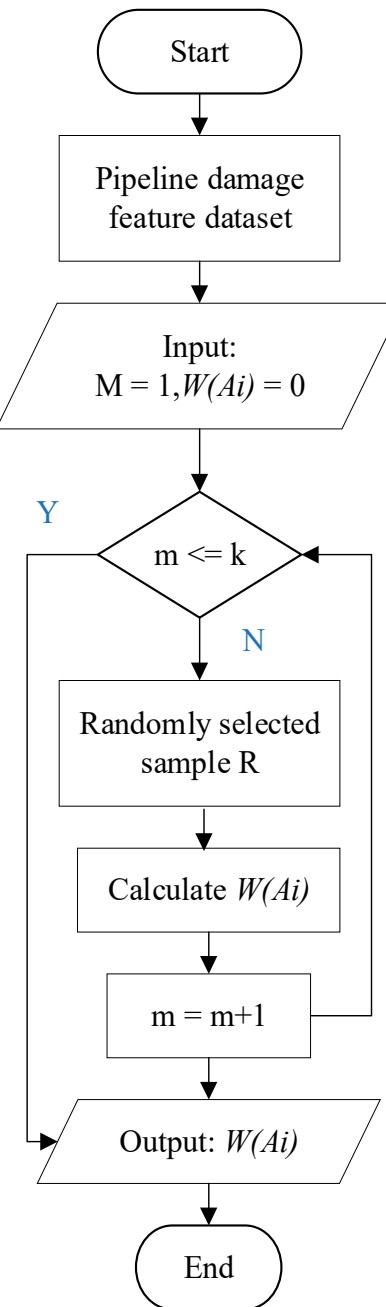

**Figure 10.** Flow chart of the feature simplification algorithm.

The process of the feature simplification-based random forest feature selection algorithm (FS-RF) is as follows: firstly, features with zero or very low weights in the pipeline damage dataset are removed using the feature simplification algorithm (FS); secondly, the random forest algorithm is used to calculate the importance of the features and rank them; finally, feature selection is performed based on the ranking results of the feature importance.

*3.2. SMA-SVM Classifier Design*

3.2.1. Support Vector Machine Principles

The SVM algorithm is a statistical machine learning technology. It uses structural risk minimization approximation to solve binary and multi-classification problems and has good applications in the case of insufficient sample size and nonlinearity [28].

Figure 11 is the schematic diagram of SVM.

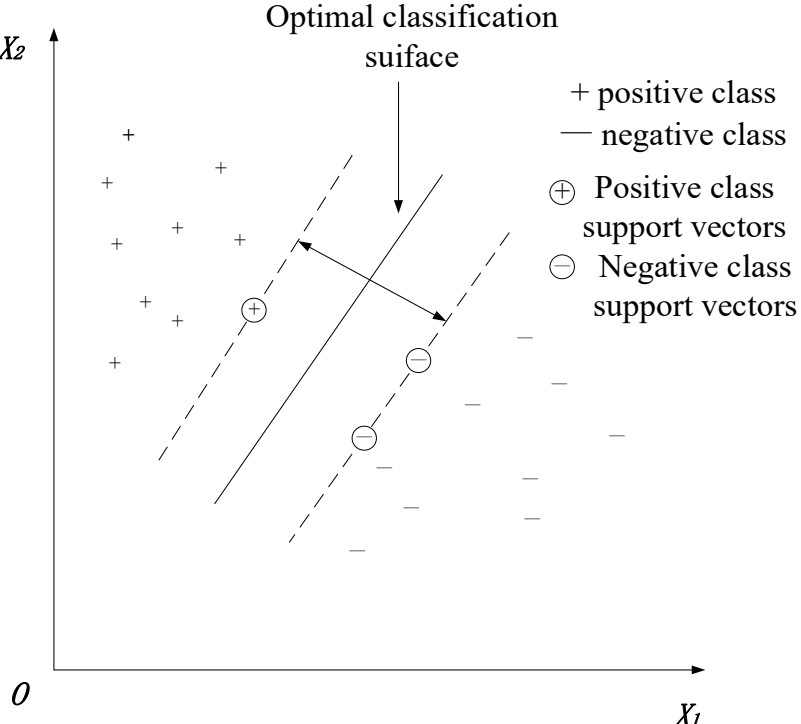

**Figure 11.** SVM schematic.

In the sample space, the hyperplane function used for classification is denoted as

$$\boldsymbol{w}^T \boldsymbol{x} + b = 0 \tag{5}$$

where $\boldsymbol{w} = (w_1; w_2; \dots; w_d)$ is the weight vector, and $b$ is the bias, for $(x_i, y_i) \in D$ if the hyperplane correctly classifies the sample, then we have

$$\begin{cases} \boldsymbol{w}^T x_i + b > 0, y_i = +1 \\ \boldsymbol{w}^T x_i + b < 0, y_i = -1 \end{cases} \tag{6}$$

The sample data satisfying Equation (4) is the "support vector," and the interval (Margin) between the two categories is defined in Equation (5) and is called the maximum interval.

$$\text{Margin} = \frac{2}{\|\boldsymbol{w}\|} \tag{7}$$

The separating hyperplane with "maximum spacing" is the one that finds the constraint parameters $\boldsymbol{w}$ $b$ in Equation (5) such that Margin it is maximum, i.e.,

$$\begin{aligned} &\max_{\boldsymbol{w},b} \frac{2}{\|\boldsymbol{w}\|} \\ &s.t. y_i (\boldsymbol{w}^T x_i + b) \geq 1, i = 1, 2, \dots, m \end{aligned} \tag{8}$$

Support vector machines use a non-linear transformation to transform the input space to a higher dimensional space; this transformation is achieved using a kernel function, the RBF kernel function has been chosen for this paper, and the formula is

$$\kappa(x_i, x_j) = \exp(-\frac{\|x_i - x_j\|^2}{2\sigma^2}) \tag{9}$$

### 3.2.2. Slime Mold Optimization Algorithm

This paper proposes using the Slime Mold Optimization Algorithm (SMA) to optimize the parameters of the SVM classification model $C$ $g$. SMA is a powerful population optimization algorithm based on the natural mucus oscillation pattern [29]. It has the advantages of strong convergence, less parameter adjustment, and easy operation. It can ensure the balance between local and global search and meet this paper's requirements for optimizing the internal parameters of support vector machines.

A mathematical model of the mucilage foraging process is developed, and the equation for an individual mucilage position update is

$$X(t+1) = \begin{cases} rand(UB - LB) + LB, & rand < z \\ X_b(t) + vb \times (W \times X_A(t) - X_B(t)), & r < p \\ vc \times X(t), & r \geq p \end{cases} \tag{10}$$

where $UB, LB$ denotes the upper and lower boundaries of the search area $rand$ is a random number uniformly distributed between the intervals $[0, 1]$. $z$ is a custom parameter (usually 0.03), $vb$ is a random number between $[-a, a]$, and $vc$ is a linear convergence from 1 to 0. $t$ denotes the current number of iterations, $X_b(t)$ denotes the current position of the best-adapted individual, $X(t)$ denotes the current position of the slime individual, $X_A$ and $X_B$ denote the positions of two randomly selected slime individuals, respectively, and $W$ denotes the weight factor of the slime indicates the weight factor of the mucilage.

The control parameter $p$ is updated with the following formula

$$p = \tanh|S(i) - DF| \quad i \in 1, 2, \cdots, n \tag{11}$$

where $S(i)$ represents the fitness value of $X$ and $DF$ represents the best fitness value in all iterations.

The interval for the parameter $vb$ is $[-a, a]$, and the function expression for $a$ is

$$a = \text{arctan}h(-(\frac{t}{T}) + 1) \tag{12}$$

where $T$ indicates the maximum number of iterations.

The updated formula for

$$W(SmellIndex(i)) = \begin{cases} 1 + r \cdot \log(\frac{bF - S(i)}{bF - \omega F} + 1), & condition \\ 1 - r \cdot \log(\frac{bF - S(i)}{bF - \omega F} + 1), & others \end{cases} \tag{13}$$

$$SmellerIndex = sort(S) \tag{14}$$

where *condition* represents the top half of the population in terms of fitness, *others* represents the remaining individuals, $r$ represents the random number of individuals in the $[0, 1]$ interval, $bF$ represents the best fitness obtained for the current number of iterations, $\omega F$ represents the worst fitness obtained for the current number of iterations, and *SmellIndex* represents the sequence of fitnesses (increasing sequence in the minimum value problem).

The parameter $vc$ takes a value between the interval $[0, 1]$ and eventually converges to 0. The updated formula is

$$vc = [-b, b], b = 1 - \frac{t}{T} \tag{15}$$

### 3.2.3. SMA-SVM Classification Model

The penalty factor $C$ in the support vector machine (SVM) classification model with the kernel function parameters $g$ was used for the optimization search using the slime mold algorithm (SMA). The classification recognition rate of 30% (397) of the test samples in the sample feature dataset was used as the value of the fitness function in the SMA algorithm

for the optimization search of the SVM parameters. The iterative steps of the SMA-SVM learning algorithm are as follows: Figure 12 is the algorithm flow chart.

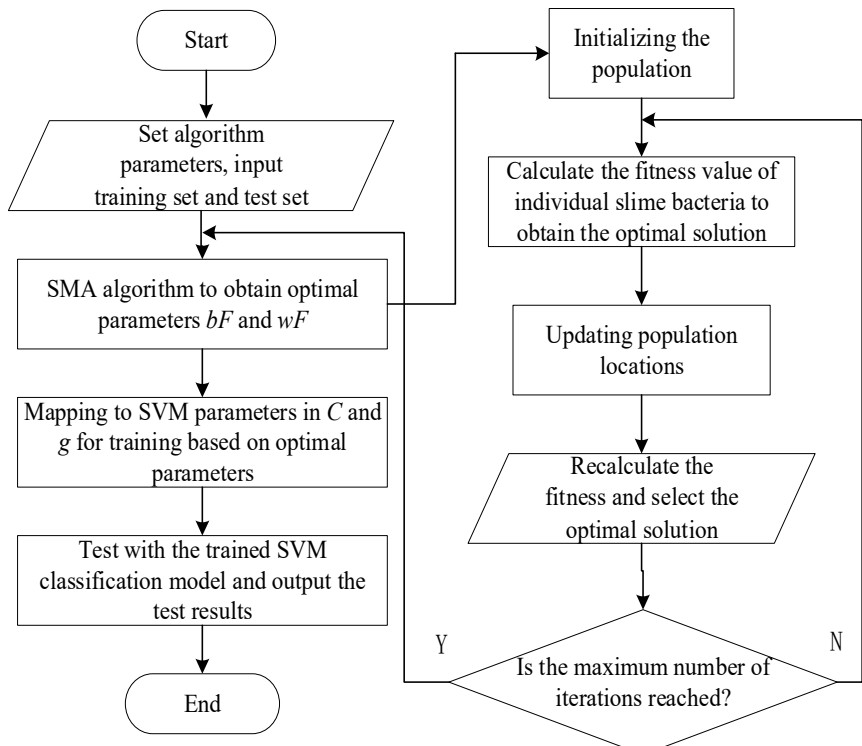

**Figure 12.** Flow chart of SMA-SVM algorithm.

Step 1    Numerical initialization, setting the relevant parameters of the SMA, such as the number of populations, the maximum number of iterations, the number of optimization parameters, the upper and lower bounds for the values of $C$ $g$, etc.

Step 2    Initialize the slime population $N$, and randomize the initial population location.

Step 3    Use the SVM classification model to calculate the fitness of each slime $fitness$ and rank the slime with the smallest fitness $samll fitness$ as the target location $best fitness$.

Step 4    Update the optimal position of the slime bacteria, as per Equation (8).

Step 5    Determine whether the maximum number of iterations has been reached. If so, continue with Step 6; otherwise, jump to Step 3 to continue the execution.

Step 6    Output the optimal parameters $(bestc, bestg)$ and map them to the SVM parameters $(C, g)$ to obtain the initial SVM model, then train the SVM model and test the SVM model.

## 4. Experiments

### 4.1. Experimental Environment Platform

The experimental environment of this study is Inter Core i5-4200M CPU 2.5 GHz, using Matlab R2016a platform and Libsvm toolbox.

### 4.2. Feature-Simplified Random Forest Algorithm

In this paper, the features of the traditional random forest algorithm and the improved random forest algorithm are ranked in importance by two metrics, Mean Decrease Accuracy (MDA) and Mean Decrease Gini (MDGini) [30], as shown in Figures 13 and 14.

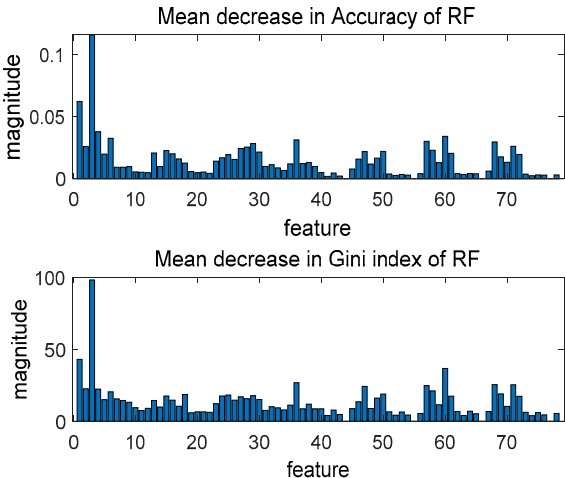

**Figure 13.** Traditional random forest feature algorithm.

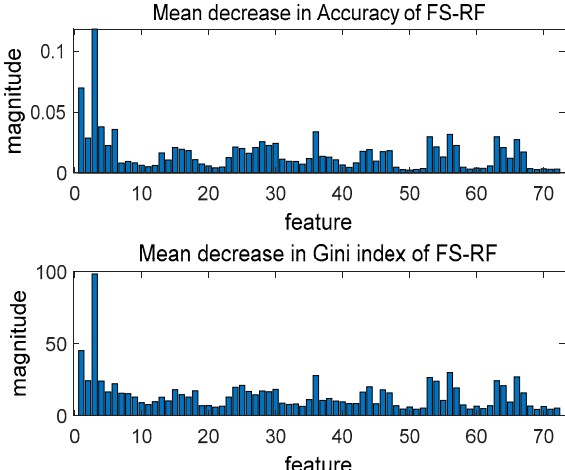

**Figure 14.** Improved random forest feature selection algorithm.

In Figures 13 and 14, "MDA" indicates the degree to which the prediction accuracy of the RF algorithm decreases. The higher the value, the more critical the function is. "MDGini" indicates the degree of influence of each variable on the heterogeneity of observations at each node of the classification tree. The higher the value, the greater the importance of the variable. When calculating feature importance, the improved random forest algorithm has used the simplified algorithm (FS) to eliminate features with zero or very low weights. Only the remaining features are analyzed for feature importance (in this paper, the features eliminated are 41, 43, 44, 55, 66, and 77); the features screened out by the improved random forest algorithm are the same as the features with the lowest importance in the traditional algorithm's importance ranking. The improved random forest algorithm effectively reduces the random forest error's upper bound and improves the feature selection's feasibility.

In Table 4, 1–78 represent the original feature dataset, A1–A78 represent the results of ranking the feature parameters of the traditional random forest algorithm, and B1–B78 represent the results of ranking the feature parameters of the random forest improved by the feature simplification algorithm. The bolded and italicized features in the table are the features eliminated by the feature simplification algorithm.

**Table 4.** The sequence of pipeline damage characteristics before and after the assessment.

| 1~78: Ranking of Original Features; A Ranking of RF Feature Evaluation Results; B: Ranking of Improved RF Feature Evaluation Results | | | | | | | | | | | | | | |
|---|---|---|---|---|---|---|---|---|---|---|---|---|---|---|
| **1** | **2** | **3** | **4** | **5** | **6** | **7** | **8** | **9** | **10** | **11** | **12** | **13** | **14** | **15** |
| A3 | A1 | A60 | A4 | A57 | A29 | A36 | A68 | A6 | A58 | A2 | A61 | A28 | A71 | A30 |
| B3 | B1 | B4 | B60 | B6 | B36 | B57 | B68 | B29 | B71 | B2 | B28 | B27 | B58 | B15 |
| 16 | 17 | 18 | 19 | 20 | 21 | 22 | 23 | 24 | 25 | 26 | 27 | 28 | 29 | 30 |
| A27 | A15 | A25 | A69 | A50 | A72 | A13 | A47 | A16 | A24 | A17 | A5 | A49 | A46 | A70 |
| B50 | B47 | B30 | B13 | B61 | B16 | B5 | B72 | B25 | B69 | B24 | B49 | B17 | B46 | B26 |
| 31 | 32 | 33 | 34 | 35 | 36 | 37 | 38 | 39 | 40 | 41 | 42 | 43 | 44 | 45 |
| A23 | A48 | A26 | A18 | A32 | A33 | A39 | A9 | A35 | A38 | A8 | A37 | A14 | A59 | A7 |
| B23 | B70 | B38 | B59 | B18 | B37 | B35 | B48 | B32 | B39 | B31 | B14 | B9 | B7 | B8 |
| 46 | 47 | 48 | 49 | 50 | 51 | 52 | 53 | 54 | 55 | 56 | 57 | 58 | 59 | 60 |
| A31 | A34 | A45 | A19 | A12 | A67 | A11 | A22 | A10 | A64 | A21 | A42 | A20 | A40 | A56 |
| B33 | B45 | B34 | B67 | B19 | B10 | B21 | B11 | B40 | B20 | B12 | B42 | B22 | B64 | B56 |
| 61 | 62 | 63 | 64 | 65 | 66 | 67 | 68 | 69 | 70 | 71 | 72 | 73 | 74 | 75 |
| A65 | A62 | A78 | A51 | A76 | A52 | A53 | A75 | A73 | A54 | A74 | A63 | A43 | A41 | A44 |
| B62 | B65 | B51 | B73 | B53 | B63 | B78 | B75 | B76 | B54 | B52 | B74 | *B43* | *B41* | *B44* |
| 76 | 77 | 78 | | | | | | | | | | | | |
| A55 | A66 | A77 | | | | | | | | | | | | |
| *B5* | *B66* | *B77* | | | | | | | | | | | | |

Through a large number of comparison experiments, the top 70% feature attributes in the feature importance ranking of the improved random forest algorithm were finally selected in this paper as the vector set for constructing subsequent feature recognition, namely 1, 2, ..., 55, with a total feature importance percentage of 94%.

### 4.3. Experimental Parameter Setting

Parameter setting of SVM: the most widely used RBF kernel function is used as the kernel function.

SMA parameter settings: the initial population size is set to 20, and the number of terminating generations is set to 200; the penalty parameter $C$ is set to 0.01 to 500; the kernel parameter $g$ is set to 0.01 to 100, and the weighting factor is set to 1. Figure 15 shows the changing trend of the fitness function value.

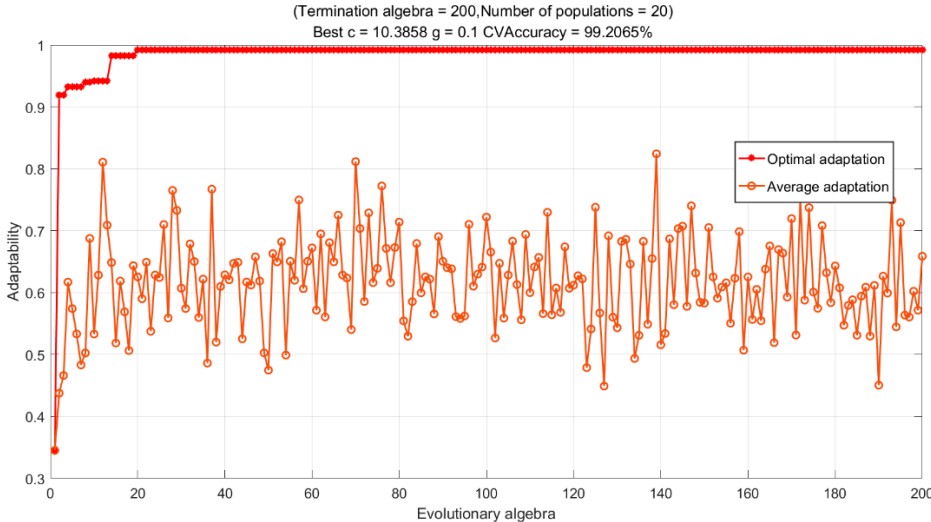

**Figure 15.** The evolutionary curve of fitness function values.

As seen in Figure 15, the penalty parameter $C$ has a value of 10.3858, and the kernel parameter $g$ has a value of 0.1 for the SVM classification model after optimization by the vicious bacteria optimization algorithm.

### 4.4. Classification Results of the FS-RF-SMA-SVM Model

Sample sets after image feature selection are divided into training and test sets and classified in the SMA-SVM classification model. At the same time, in order to better demonstrate the recognition ability of the newly constructed Support Vector Machine Classification Model (SMA-SVM) optimized by the Myxobacteria Optimization Algorithm for corrosion detection in water supply pipelines, its performance is compared with the traditional SVM classifier, the Support Vector Machine Classification Model (DFP-SVM) optimized by differential pollination in [13], and so on. The support vector machine classification model (PSO-SVM) of [31] particle population optimization and the BP network of [32] were compared. These benchmark models were selected because they have been proven by previous studies to be a method for pattern classification. The confusion matrix graph of classification and recognition results is shown in Figures 16–25 (where RF represents a random forest algorithm feature and FS-RF represents an improved random forest algorithm).

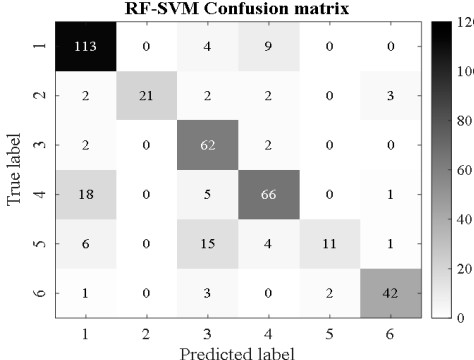

**Figure 16.** RF-SVM.

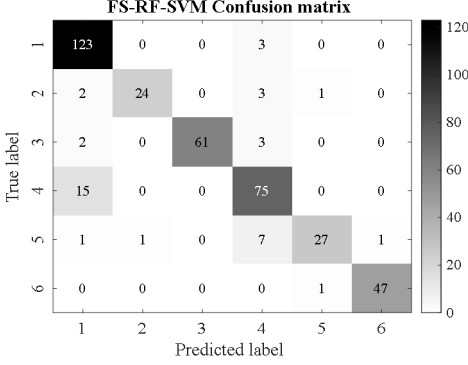

**Figure 17.** FS-RF-SVM.

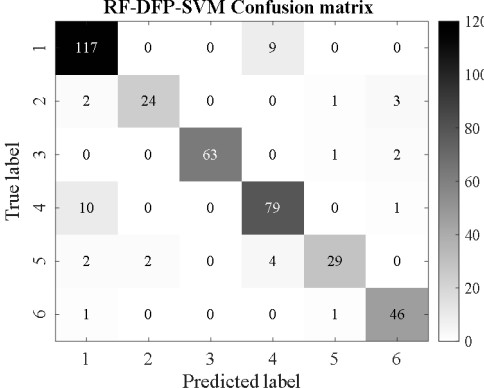

**Figure 18.** RF-DFP-SVM.

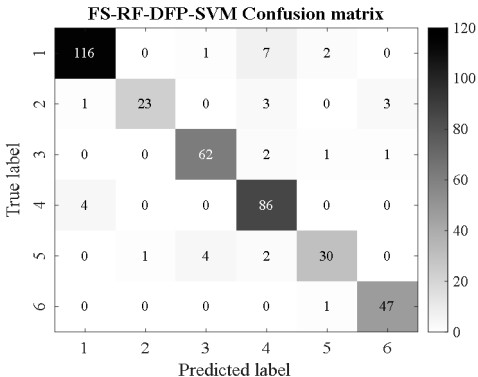

**Figure 19.** FS-RF-DFP-SVM.

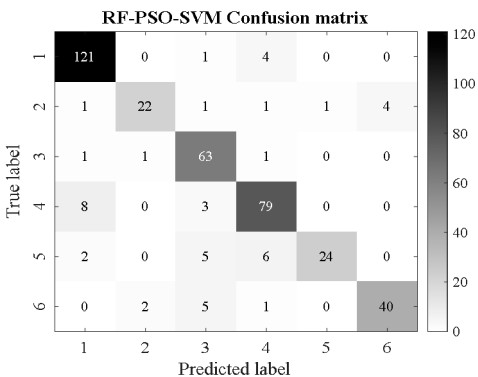

**Figure 20.** RF-PSO-SVM.

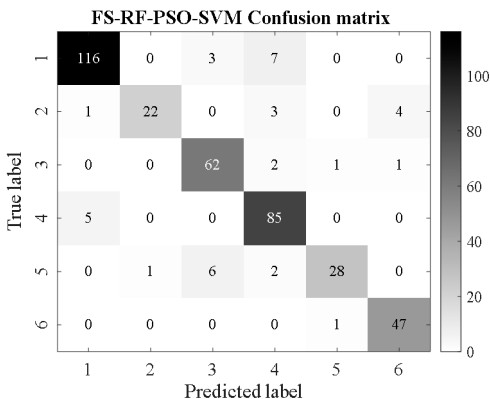

**Figure 21.** FS-RF-PSO-SVM.

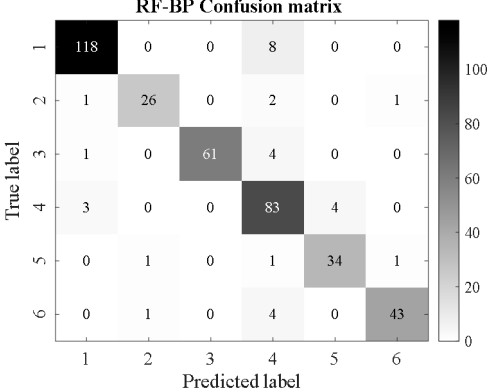

**Figure 22.** RF-BP.

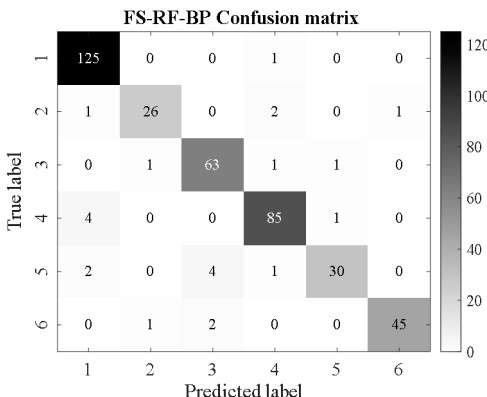

**Figure 23.** FS-RF-BP.

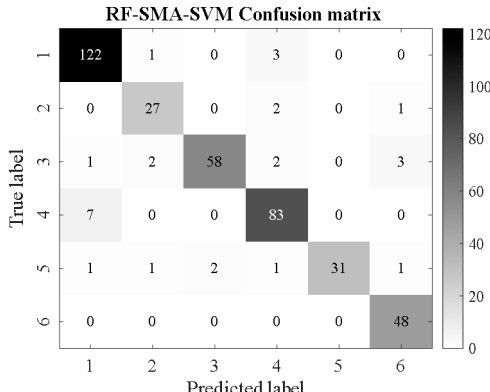

**Figure 24.** RF-SMA-SVM.

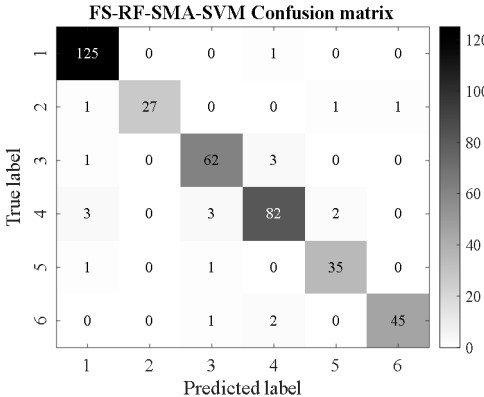

**Figure 25.** FS-RF-SMA-SVM.

From Figures 16–25, it can be seen that for the five classifier models, the confusion matrix graph of classification results shows that the number of correct samples for the improved random forest algorithm feature selection is more than that for the random forest algorithm. For the confusion matrix graph of classification results of the same kind of feature data, it can be reflected that the correct number of samples classified by the SMA-SVM classification model is more than that classified by the traditional SVM classifier. The literature [13] proposed a support vector machine classification model optimized by differential flower pollination (DFP-SVM), and the literature [31] proposed a support vector machine classification model with particle population optimization (PSO-SVM), and the literature [32], BP network classification model, from which the correct number of samples can be obtained. In this paper, the random forest algorithm improved by the feature simplification algorithm and the SVM classification model improved by the myxobacteria

optimization algorithm can improve the accuracy of identifying the characteristics of the damage image on the inner wall of pipelines.

The image characteristics of six different types of pipeline wall damage samples are compared under the improved random forest algorithm and the improved SVM classification model. As shown in Table 5, the number of normal pipeline samples in the test set is 126, the number of color-order corroded pipeline samples is 66, the number of texture-order corroded pipeline samples is 90, the number of point-type corroded pipeline samples is 30, and the number of local corroded pipeline samples is 37. There are 48 global corrosion pipeline samples and 397 total test set samples.

**Table 5.** Comparison of test classification results for the six types of pipe damage sample sets.

| Pipeline Category | SVM | Literature [13] | Literature [31] | Literature [32] | Algorithms in This Paper |
|---|---|---|---|---|---|
| Normal pipeline (126) | 123 (97.2%) | 116 (92.1%) | 116 (92.1%) | 125 (99.21%) | **125 (99.21%)** |
| Color order corrosion (66) | 61 (92.42%) | 62 (93.94%) | 62 (93.94%) | 63 (95.45%) | 62 (93.94%) |
| Texture order corrosion (90) | 75 (83.33%) | 86 (95.55%) | 85 (94.44%) | 85 (94.44%) | 82 (91.11%) |
| Pitting corrosion (30) | 24 (80.00%) | 23 (76.67%) | 22 (73.33%) | 26 (86.67%) | **27 (90.00%)** |
| Localized corrosion (37) | 27 (72.97%) | 30 (81.08%) | 28 (75.67%) | 30 (81.08%) | **35 (94.59%)** |
| Global corrosion (48) | 47 (97.92%) | 47 (97.92%) | 47 (97.92%) | 45 (93.75%) | 45 (93.75%) |
| Total number of correct (397) | 357 | 364 | 360 | 374 | **376** |
| Classification accuracy | 89.924% | 91.687% | 90.680% | 94.207% | **94.710%** |

For normal pipeline, pitting pipeline, and locally corroded pipeline, the classification result of this algorithm is the best, with accuracy of 99.21%, 90.00%, and 94.59%, respectively. Although the algorithm is not optimal for color-order corrosion, texture-order corrosion, and global corrosion pipelines, there is little difference between the classification results and the optimal algorithm. Therefore, from the viewpoint of the classification results of individual pipe damage image categories, the classification results of the above algorithms are not significantly different. Still, from the overall classification results, the classification results of this algorithm are the best and are relatively stable, with 376 correct samples classified—an accuracy rate of 94.710%. Therefore, in summary, this improved algorithm's recognition and classification results are better than other classification algorithms and have high generalizability.

Next, the overall performance of several classification models after feature selection of the traditional random forest algorithm and the improved random forest algorithm was analyzed in terms of algorithm accuracy (Accuracy), precision (Precision), recall (Recall), F1-score and mean square error (RMSE.) [33]. Table 6 shows the results, and to show more graphically the change curves of the classification results of the improved algorithm in this paper with those of the traditional algorithm and other optimization algorithms under these evaluation parameters, Figure 12 compares the results using a bar chart.

**Table 6.** Comparison of test classification results for the pipeline damage sample set.

| Parametric Algorithms | | Precision | Recall | F1-Score | RMSE. | Accuracy |
|---|---|---|---|---|---|---|
| RF | SVM | 0.835 | 0.740 | 0.785 | 16.093 | 79.345% |
| | Literature 13 | 0.910 | 0.884 | 0.897 | 7.245 | 90.176% |
| | Literature 31 | 0.882 | 0.837 | 0.859 | 8.631 | 88.161% |
| | Literature 32 | 0.925 | 0.911 | 0.918 | 5.598 | 91.940% |
| | SMA-SVM | 0.931 | 0.918 | 0.924 | 5.385 | 92.947% |
| FS-RF | SVM | 0.925 | 0.874 | 0.899 | 8.124 | 89.924% |
| | Literature 13 | 0.918 | 0.895 | 0.906 | 6.205 | 91.688% |
| | Literature 31 | 0.913 | 0.879 | 0.896 | 6.916 | 90.680% |
| | Literature 32 | 0.941 | 0.918 | 0.929 | 4.103 | 94.207% |
| | SMA-SVM | 0.954 | 0.937 | 0.945 | 4.143 | **94.710%** |

From the experimental results in Table 6 and Figure 26, it can be seen that by comparing the recognition and classification results of the BP neural network, SVM, and optimized SVM classification algorithm models, the accuracy, recall, F1 score, and accuracy of the algorithm proposed in this paper are higher than those of other algorithms, and the mean square error index value also has good results. Therefore, the improved classification algorithm in this paper has a good classification effect and practicability. At the same time, by comparing the classification results of the improved random forest algorithm and the random forest algorithm, it can be seen that the values of the five evaluation indicators of the improved RF classification results are better than the RF classification results, which verifies the effectiveness of the improved feature selection algorithm in this paper. Combined with Table 6, it can be concluded that the SVM classification model optimized by SMA has better classification results for normal pipes, pitting corrosion, and locally corroded pipes and the classification results for other pipes are less different from its optimal algorithm. In summary, the analysis can be concluded that the FS-RF-SMA-SVM model algorithm can provide technical support for pipe damage detection.

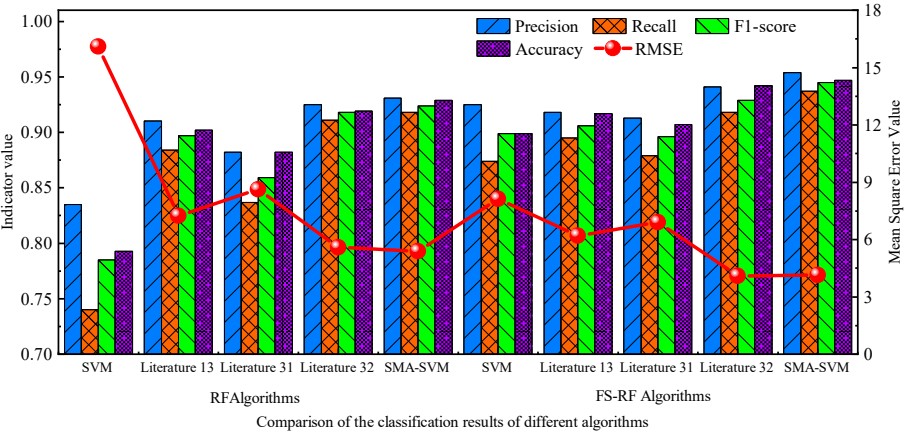

**Figure 26.** Comparison of test classification results for the pipeline damage sample set.

## 5. Conclusions

This study first proposes a feature selection random forest algorithm based on feature simplification, which solves the problem of the reliability of attribute weights when traditional random forest algorithms partition more feature data, considers the influence of correlation between feature variables on recognition accuracy and reduces the influence of redundant features on the algorithm. Then slime mold algorithm is used to optimize the kernel function parameters and penalty factors of the SVM model. Finally, the proposed model is applied to the classification and prediction of pipeline corrosion damage data sets. Experimental results show that the classification accuracy of the SMA-SVM algorithm based on FS-RF feature selection proposed in this paper is better than other literature algorithms. Test samples (399) were divided into 376 pairs, and the accuracy was 94.710%, 4.786%, 3.023%, 4.03%, and 0.503% higher than that of traditional SVM, DFP-SVM, PSO-SVM, and BP neural network, respectively. The experimental results meet the expected requirements, which provides a new idea for the damage detection of the inner wall of the water supply pipeline.

However, with the development of society and the increase in market demand, the requirements for pipeline detection technology will become higher and higher in the future. Therefore, the work of this paper still needs to be improved. In future research, the following aspects can be strengthened.

(1)   In terms of feature dimensionality reduction, this paper uses an improvement of the traditional random forest algorithm, which has good results for the feature data in this paper. However, the classification effect on the new feature data set still needs to be

studied; therefore, further improving the generality of the algorithm and overcoming the limitations of the feature data are the key points to be learned in the future.

(2)    In terms of research objects, this paper only studied the common damage (corrosion) on the inner wall of the pipeline, and further research is needed to identify other damage categories, such as pipeline cracks, pipeline fractures, etc.

(3)    From the aspect of damage identification and classification, the popular depth learning technology can be used to realize the identification of pipe wall damage, and further improve the accuracy of identification.

**Author Contributions:** Conceptualization, Q.Z. and L.L.; methodology, L.L.; software, L.Z. and M.Z.; validation, Q.Z., L.L. and L.Z.; investigation, L.Z. and M.Z.; writing—original draft preparation, L.L.; writing—review and editing, Q.Z. and M.Z.; visualization, L.L. and L.Z.; supervision, Q.Z. All authors have read and agreed to the published version of the manuscript.

**Funding:** This research was supported by the National Natural Science Foundation of China (Grant No. 51804248), the Shaanxi Provincial Science and Technology Department Industrial Research Project (Grant No. 2022GY-115), the Beilin District Applied Technology R&D Project (Grant No. GX2114) and the Shaanxi Provincial Education Department Service to Local Enterprises (No. 22JC050).

**Institutional Review Board Statement:** Not applicable.

**Informed Consent Statement:** Not applicable.

**Data Availability Statement:** Data is available on request from the corresponding author.

**Conflicts of Interest:** The authors declare no conflict of interest.

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
