# Peer review of "Recognition of Corrosion State of Water Pipe Inner Wall Based on SMA-SVM under RF Feature Selection"

_coatings, doi:10.3390/coatings13010026_

Round 1

Reviewer 1 Report

Dear Authors

the paper is very interesting, however to improve interest to the readers you should consider in your references about corrosion of dental materials, in particular titanium and ceramics. Please cite PubMed ID35564533, DOI10.23805/JO.2018.10.04.04, DOI10.3390/app12136729, and DOI10.3390/ijerph191811735

The strenghts of the paper are related to the proposed model and statical analysis, very accurate, that bring results very interesting for the readers.

Indeed the forest algorhytm is very significant for the presented results.

Reviewer 2 Report

Recognition of Corrosion State of Water Pipe Inner Wall Based on Support Vector Machine of Slime Mold Optimization Algorithm under Random Forest Feature Selection

The paper presented interesting research regarding the application of slime mold optimization to increase the accuracy of the pipeline internal wall damage detection. I noted several components which need attention to be revised and clarified.

1. Title is too long. Make it comprehensive.

2. Literature review is done in the draft, but no milestone study is found.

It is necessary to support your state-of-the-art.

3. It is mentioned in the draft that the experiment is conducted. However, I cannot find any configuration and setup descriptions.

4. Related to no 4, if the proposed algorithm processes the data, what is the data source used in this work? Is it experimental data? or measurement of the actual case? If they are taken from other literature, what makes the data qualified or good enough to be input for this calculation? Clarification regarding this has not clearly written.

5. As a follow-up, the methodology must be clearer. Step-by-step details, including input and boundary conditions, must be provided.

6. Check lines 146 and 147. They must be placed on the next page. 

7. Mean Decrease Accuracy (MDA) and Mean Decrease Gini (MDGini) can be given citations.

8. Give recommendations for future works in Conclusions based on your findings.

9. Check author guidelines for references. They need correction and re-formating.

Reviewer 3 Report

1. Qualitative information’s are missing in abstract. Abstract should be to improve with more specific results.

2. The introduction has been formulated; recent bibliographic references have been added

3- In the first time you must use the full name and after add their abbreviation as an example SVM Line 64

4. The author should compare and justify, how these result are better than the already reported in literatures.

5.  Why choosing Cast iron for this study? Please mention the industrial application of this across the different industry.

6. Figures 1, 2, 3 and 4 are not clear, make the writings come out of these images

7.  in figure 4 the author talks about different types of corrosion, what are the criteria used to identify these criteria (Insufficient naked eye)

8.  It is important for authors to add some advanced surface study such as AFM and XPS before and after test corrosion

9.  In line 328, you speak of accuracy of 99.206%, justify this with a reference

10.  The conclusion paragraph was highly modified

11.  Update and add new bibliographic references

12. if possible and to enhance this work add electrochemical tests against corrosion such as electrochemical impedance spectroscopy

Round 2

Reviewer 2 Report

All comments have been well addressed in the revision stage.

Reviewer 3 Report

no comment